# Position: Explainability Research Must Prioritize Foundations over Ad-hoc Methods

**Michal Moshkovitz** [* 1]  **Suraj Srinivas** [* 2]  **Lesia Semenova** [* 3]  **Nave Frost** [4]  **Cyrus Rashtchian** [5]
**Valentyn Boreiko** [6]  **Shichang Zhang** [7]  **Himabindu Lakkaraju** [7 5]  **Cynthia Rudin** [8]  **Jennifer Wortman Vaughan** [9]

## Abstract

Despite the proliferation of Explainable AI (XAI) techniques—from feature attributions to sparse autoencoders—explanations rarely influence real-world workflows. In practice, they are often generated and discarded without guiding meaningful action. This gap reflects foundational shortcomings: research has not yet established methodologies for integrating explanations into end-to-end, human-in-the-loop systems. This position paper argues that the machine learning community must pivot from ad-hoc XAI methods toward addressing foundational & structural challenges, including unclear problem formulations, underspecified evaluation objectives, and the absence of pipelines for explanation-driven feedback. We support this claim through an analysis of recent ICML, NeurIPS, and ICLR papers and a survey of XAI practitioners, revealing recurring issues that limit cumulative progress. We conclude by outlining a practical checklist designed to shift XAI toward a more human-centered, action-oriented paradigm. By emphasizing foundational clarity over the development of ad-hoc methods, we hope to provide a roadmap for integrating explanations into actionable, feedback-driven AI systems.

## 1. Introduction

Despite impressive engineering advances, we lack a fundamental understanding of why AI models produce the outputs they do and what makes them go wrong. This opacity not only raises concerns around trust and safety, but can also slow progress towards the building of reliable AI models and applications (Amodei, 2025). Research on *explainable AI* (XAI) aims to provide methods and tools that help model developers, end users, and other stakeholders understand how a model behaves and why it produces its outputs (Doshi-Velez & Kim, 2017).

The XAI literature has produced a wide range of post-hoc techniques to probe different aspects of model behavior, including feature attribution (Ribeiro et al., 2016; Lundberg & Lee, 2017), data attribution (Koh & Liang, 2017), counterfactual explanations (Wachter et al., 2018; Ustun et al., 2019), concept-based methods (e.g., sparse autoencoders, Bricken et al., 2023), chain-of-thought explanations for LLMs (Wei et al., 2022; Lanham et al., 2023), and analyses of internal computations (e.g., mechanistic interpretability) (Olah et al., 2020; Elhage et al., 2021). We distinguish such post-hoc *explainability* (i.e., XAI) methods from *interpretable models* — glass-box models whose decision logic is specified explicitly by design and constrained to human-scale complexity (Rudin, 2019; Caruana et al., 2015). While interpretable models are preferred when feasible (Rudin, 2019), many modern models, including LLMs and other foundation models, lack known interpretable-by-design formulations, and post-hoc methods are often the only available tools for understanding model behavior.

Despite substantial methodological diversity, XAI approaches have been shown to be unreliable, highly sensitive to arbitrary design choices, or misused in practice (Kindermans et al., 2019; Slack et al., 2020; Kaur et al., 2020; Lin et al., 2021; Wu et al., 2025). Moreover, despite extensive research activity, the impact of post-hoc explainability tools remains limited among downstream stakeholders such as model developers, domain experts, decision-makers (Bhatt et al., 2020; Casper, 2023; Smith et al., 2025). Rather than treating this gap as a motivation for increased methods development, we argue that it reflects deeper foundational &

---
[1]Google Research, Israel [2]Bosch Center for Artificial Intelligence & Bosch Research, Sunnyvale, USA [3]Rutgers University, New Brunswick, NJ, USA [4]eBay Research, Israel [5]Google Research, USA [6]Amazon, Germany [7]Harvard University, USA [8]Duke University, USA [9]Microsoft Research NYC, USA. Correspondence to: Michal Moshkovitz <michal.moshkovitz@mail.huji.ac.il>, Suraj Srinivas <suuraj.srinivas@gmail.com>, Lesia Semenova <lesia.semenova@rutgers.edu>.

*Proceedings of the 43rd International Conference on Machine Learning*, Seoul, South Korea. PMLR 306, 2026. Copyright 2026 by the author(s).

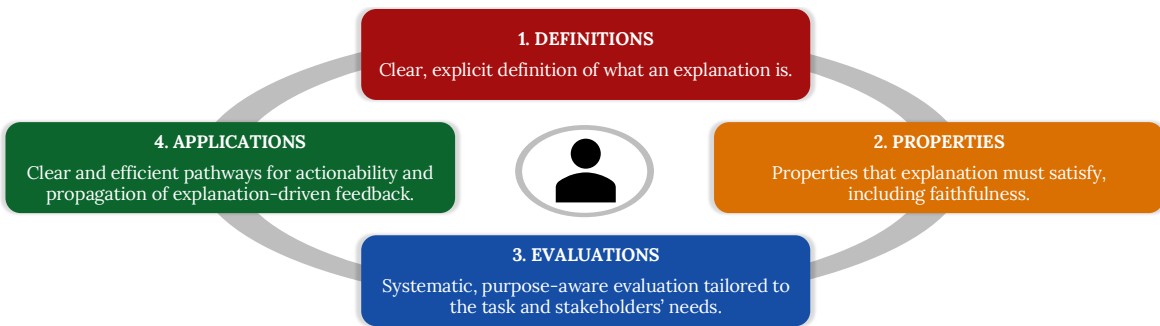

*Figure 1.* We lay out four interdependent challenges centered on definitions, properties, evaluations, and applications. Stakeholders' needs and constraints shape how each challenge is instantiated, guiding the design, evaluation, and deployment of explanations. Because these challenges are deeply interconnected, progress on one challenge requires and supports progress in the others.

structural issues — specifically, the lack of explicit explanatory objectives, falsifiable claims, principled evaluation criteria linked to stakeholder needs, and practical interactive pipelines. More broadly, the field has largely inverted the natural research order: developing methods before establishing clear objectives and evaluation criteria. This inversion makes it difficult for results to be meaningfully accumulated, compared, and assessed. In this position paper, we call for a **re-prioritization of machine learning research in explainability to address fundamental questions such as its definition and utility, with a de-prioritization of ad-hoc methods development.** We do not argue for a pause on explainability methods development, nor do we deny that practitioners may find utility with these tools. Rather, we argue that methods development has outpaced the foundations needed to integrate them into reliable workflows. Our critique is aimed primarily at methods that claim explanatory utility for stakeholders, not at research on model internals whose primary goal is to advance scientific understanding of deep learning systems.

This paper views explainability not as a property of the model alone, but as a fundamentally human-centric. Explanations are only valuable if they are understandable and actionable to the stakeholders who interact with them, and different stakeholders have different needs. We echo the calls from the human-centered XAI community (Vaughan & Wallach, 2021; Liao et al., 2020; Liao & Varshney, 2021; Ehsan et al., 2022) and argue it is essential to place humans at the center of explainability research. However, within the machine learning (ML) community, these considerations are often left under-specified or deferred. Without clarifying who they are meant to serve and for what aim, explanations risk becoming detached from real-world utility. For instance, a method may successfully visualize internal model representations yet fail to guide a stakeholder's downstream decisions or actions. In this sense, human-centric considerations are not auxiliary, but central to defining and achieving explainability.

To make progress, we argue that explainability research must confront several foundational challenges that arise at different but tightly interconnected levels (see Figure 1). These challenges span the definitional level, which concerns how explainability goals and claims are articulated and what properties explanations are expected to satisfy; the evaluation level, which concerns how those properties and claims are operationalized and tested; and the deployment level, which concerns how explanations are used in practice, interpreted by stakeholders, and refined over time. Importantly, properties such as faithfulness or robustness are not linked to a single level: they must be defined precisely, evaluated rigorously, and interpreted in context. Although these challenges can be described separately, they are deeply interconnected and cannot be addressed in isolation.

The ordering of these challenges reflects a logical dependency rather than a research workflow. In practice, researchers may enter the problem from any point — for example, by identifying a desired property such as actionability, or by encountering a concrete application need that motivates explanation. We do not view explainability research as proceeding in a fixed sequence; instead, progress at any entry point ultimately requires alignment across challenges. Applications expose missing definitions; properties demand operational evaluation; and evaluation failures often reflect ill-specified goals. Stakeholders' needs and constraints influence this entire loop. Without this alignment, advances at one level fail to translate into cumulative progress. We identify the following challenges:

1. **Definitions**: How can we *define explainability* in a way that reflects the intended purpose and mode of use? In other words, how can we develop rigorous definitions that are grounded in what the explanation is for (be it debugging, auditing, scientific insight, or decision support) and specify the corresponding foundational requirements and guarantees?

2. **Properties**: What properties must an explanation satisfy

to meaningfully represent model behavior? In most cases, *faithfulness* is necessary: any explanation claiming to reflect a model must be verifiably consistent with the underlying model or with the specific behavioral property it targets. However, faithfulness alone is insufficient, as explanations may be correct yet too coarse to provide meaningful insight. Explanations should therefore satisfy *additional, use-dependent properties*, such as completeness or robustness. The central challenge is to define these properties precisely and associate them with rigorous tests. Without this, explanations remain heuristics rather than reliable representations of model behavior or stakeholder goals.

3. **Evaluations**: How can we design rigorous and human-centered *evaluations* for explainability methods? While many studies call for "better evaluation" (Nauta et al., 2023; Sokol & Vogt, 2024; Löfström et al., 2022), human evaluation is still rare in explainability papers within the ML community. We argue the field should shift towards evaluating the utility of explanations for specific tasks (e.g., debugging, auditing, or decision-making) and explore a variety of user study methods, from interview studies to controlled experiments, in collaboration with researchers from human-centered fields. These studies would complement existing metrics that aim to capture desirable explanation properties, such as sparsity, stability under perturbations, computational efficiency, or agreement with the model's internal reasoning or output behavior (i.e., high fidelity).

4. **Applications**: How can explanations move beyond standalone inspection to efficiently support action and feedback in real workflows, meaningfully shaping decisions, updates, and oversight over time? This requires specifying what actions explanations enable, how feedback is interpreted, and how it propagates through the system.

Unlike prior position papers that primarily articulate desiderata (Doshi-Velez & Kim, 2017; Saphra & Wiegreffe, 2024; Sharkey et al., 2025), this work combines empirical analysis of recent XAI research with practitioner evidence (Sections 2.1–2.2) to diagnose why prior calls have not translated into cumulative progress. We argue that the core issue is not awareness, but misalignment between explanatory objectives, validation, evaluation, and deployment. Explanations are frequently treated as standalone outputs, missing the formal practice needed to support downstream action and iterative feedback. We conclude by discussing the opportunities that may emerge if the identified challenges are successfully approached (Section 4), and providing a paper checklist to guide future research on XAI.

## Alternative Views

While we have so far argued for prioritizing foundational questions in explainability, we now discuss potential drawbacks associated with such an agenda.

**Current literature may not be mature enough to establish foundations.** The rapid pace of model development and deployment requires a parallel pace in explainability research. Many existing methods are fundamentally broken or misleading, so building foundations by refining or generalizing them may be counterproductive. Instead, discarding unpromising directions and exploring new ones might be the more practical route, even if the foundational work is not yet complete. We believe such arguments view foundations development through a narrow lens: we do not see it as being limited to proving theorems about existing methods, but building conceptual frameworks (with rigorous goals and evaluations) under which future methods can operate.

**It may be faster to make short-term empirical progress.** Developing new explainability methods can yield quick, empirical insights that are immediately useful for practitioners, even if they are not rigorous. Unlike foundational work, which may take years to converge into actionable frameworks, ad-hoc method development offers iterative feedback and the potential for near-term impact. Especially in applied settings, this kind of agility is often desirable. However, neglecting long-term foundational research can be shortsighted: carefully developing core principles and theories may enable far more significant and lasting advances than any sequence of incremental method tweaks. In the long run, strong foundational understanding can guide more effective methods, unify fragmented approaches, and avoid wasted effort on dead-end directions.

**Foundational consensus can emerge from ad-hoc methods development.** In machine learning, the historical trend seems to be that ad-hoc methods precede theory. For example, in deep learning, architectures and algorithms were discovered before the underlying theory, which still under development. It may be tempting to believe the same might hold for XAI. We believe this possibility is unlikely. In deep learning, the methods had real measurable utility; predictive performance improved, benchmarks converged, and the empirical signal was clear. The situation with XAI is different, it is not just that the methods are theoretically ungrounded, it is more severe: we lack a clear empirical signal of progress. We therefore believe that continuing to develop ad-hoc methods in XAI is more likely to compound the existing confusion than to resolve it.

## 2. Understanding the State of XAI Research

This section presents findings from our efforts to understand the state of XAI research along two complementary axes.

First, we present results from an LLM-driven literature survey of XAI papers, to understand the high-level trends in topics and methodology. Second, we present findings from a survey of XAI researchers and practitioners about their usage of XAI tools, and the main challenges faced.

## 2.1. What Do ML Explainability Papers Study? An LLM-driven Literature Survey

We present findings from an LLM-driven literature survey. The purpose of this survey is to give an overview of the state of research on XAI within the ML community. We do this by having an LLM (specifically, Gemini 2.5 Flash) answer simple binary questions regarding claims made in each paper. We emphasize that we do not use the LLM to judge scientific merit or methodological soundness, but only for gathering statistics on claims made in the papers.

We analyze papers published at NeurIPS, ICML, and ICLR in 2023–2024.[1] Relevant papers are identified via a two-stage filtering process: we first screen abstracts using the keywords *explainability*, *explainable*, *interpretability*,[2] and *interpretable*, and then use an LLM to assess whether the paper primarily claims to study explainability of ML models. Altogether, 617 papers determined as relating to XAI were analyzed. Papers passing both stages are analyzed using a fixed set of 20 questions probing objectives, methods, and contributions to explainability (see Appendix for more details).

To get a sense of the accuracy of the LLM, we randomly sample 25 (research paper, question) pairs and manually answer the questions. We then compare our answers to LLM responses, obtaining an agreement of 88%. Below we present high-level insights from the LLM-driven literature survey. **1. Most papers propose novel methods:** We found that 77% of papers proposed novel methods for explainability. This suggests that explainability research is still primarily driven by the development of new methods.

**2. Few papers provide formal definitions:** We found only 11% of papers contain any formal definitions of explainability objectives, even though roughly 20% of the papers emphasized theoretical aspects. Upon closer inspection, we found that the theoretical components of these papers typically addressed issues peripheral to explainability.

**3. Evaluation focuses on faithfulness, and human evaluation remains rare:** While roughly 17% of papers proposed new evaluation metrics for explainability, only 11% included user studies. Faithfulness evaluations were by far the most common, with 56% of papers claiming to perform them. Anecdotally, machine learning researchers may be less inclined to run user studies because they are more expensive and slower to execute. We note that in human-computer interaction (HCI) venues, user studies and other human-centered forms of evaluation are more common (Lai et al., 2023), but these were not included in our survey.

**4. Computer vision applications are the most common:** While 76% of papers demonstrate or discuss a concrete downstream impact of explainability, computer vision applications were the most common at 46%, as opposed to 26% of papers demonstrating applications in NLP, and less than 1% showcasing speech or audio-related applications.

**5. Concept-based explanations and mechanistic interpretability are the most common topics:** We find 31% of papers are about concept-based explanations, including sparse autoencoders (SAEs). The second most common topic is mechanistic interpretability at 19%. This was closely followed in third place by feature attribution and saliency maps at 18%. Data attribution was the least common, with only 2% of papers on the topic. We note that these topics are not mutually exclusive, as we suspect many SAE papers are framed as mechanistic interpretability papers.

**Summary.** Our LLM-driven literature survey reveals that the field is driven primarily by method development, with limited attention to definitions, evaluation standards, or human-centered assessment. This gap underscores the need for more foundational approaches to explainability research.

## 2.2. What Practical Challenges Exist for Explainability? A Survey of Researchers and Practitioners

To explore the practical challenges of explainability in real-world settings, we conducted a survey with 34 researchers and practitioners who had direct experience with explainability methods. Information on participant demographics, such as years of experience in ML and the primary research area, is provided in Appendix B.[3] The survey asked participants to describe a recent project involving an explainability method. We focused on goals and outcomes, asking about the intended audience, the objectives of the explanations, and the extent to which these were achieved, as well as how participants evaluated correctness and effectiveness. Finally, we invited reflections on challenges faced during implementation. For many of these questions, multiple response options were provided, and participants could select more than one. Our high-level insights are as follows:

**1. Diverse goals:** The primary intended goals of explana-

---

[1]We also analyzed papers from adjacent CV and NLP venues (CVPR and ACL), finding similar high-level trends with expected domain-specific differences in applications and methodology. See Appendix A for more details.

[2]We include *interpretability* and *interpretable* for recall purposes to ensure coverage of papers that self-describe post-hoc analysis of model behavior using this terminology (e.g., mechanistic interpretability). Definitions are clarified in the Introduction.

[3]This survey was approved by the Institutional Review Board (# IRB25-0391) at Harvard University.

tions were understanding patterns in data (58.8%), enhancing trust (58.8%), and debugging models (52.9%), while only 8.8% mentioned compliance with regulations.

**2. Heavy reliance on feature attribution, but fragmented practice:** The survey revealed considerable dispersion in terms of methods used: six classes of XAI methods were reported, and even within the most popular class, feature attribution (70.6%), 16 different specific techniques were reported. This dispersion may reflect legitimate variation in tasks and use cases. At the same time, it highlights a practical difficulty: users must often choose among many methods without clear guidance about their relative strengths, limitations, and appropriate use cases.

**3. Method choice driven by ease-of-implementation over rigor:** When selecting methods, the main reasons were ease of implementation or documentation (52.9%) and popularity (47.1%), with only 29.4% citing rigorous research backing. This suggests that adoption is guided more by availability and community norms than by formal guarantees.

**4. Evaluation practices are ad-hoc and inconsistent:** Evaluation was highly varied: 44.1% compared explanations to expert knowledge, 38.2% relied on sanity checks, 35.3% used quantitative metrics, and only 11.8% conducted user studies. Alarmingly, 23.5% admitted they either did not evaluate faithfulness or assumed it based on expectations. This is in line with the lack of systematic evaluation frameworks and parallels the finding that only a small fraction of explainability papers propose robust evaluation methods.

**5. Key pain points:** When asked about the challenges they faced while using XAI methods in practice, two challenges stood out, each cited by 44.1% of respondents: (1) Not knowing how to check explanation quality. This reflects the broader issue of lacking systematic evaluation frameworks. (2) Difficulty explaining results to end users. This difficulty arises from the lack of stakeholder-centered objectives: explanations are produced without clear formalization about who they are for or how they should be interpreted, leaving practitioners without guidance. Other significant issues included integration into workflows (38.2%) and unclear parameter choices (26.5%).

**6. Optimism about the value of explanations:** A majority of respondents reported that XAI methods were beneficial: 44.1% said they helped significantly and another 32.4% found them moderately helpful. They viewed explainability methods as supporting tasks such as data understanding and debugging, even in the absence of clear definitions, systematic evaluation, or integration into workflows. This is reflected by 44.1% of practitioners reporting not knowing how to check explanation quality, and 38.2% lacking means to integrate explanations into their workflows. Thus, these benefits are often achieved through informal, ad hoc use

rather than principled or efficient processes, limiting their reliability and transferability. This gap highlights substantial unrealized potential: addressing the foundational challenges identified in this paper would enable explainability methods to function more systematically and at scale. However, we note two caveats. First, as with any survey, we acknowledge possible selection bias, as practitioners who found explanations useful may have been more inclined to participate. Second, prior work has shown that in some cases, practitioners express enthusiasm for explanations even when they misunderstand and misuse them (Kaur et al., 2020).

**Summary.** Overall, our survey of researchers and practitioners reveals that the main bottlenecks to effective deployment are not a lack of need among practitioners, nor lack of available methods, but a lack of fundamentals. The community prioritizes availability and ease-of-use over theoretical soundness, leading to an environment where evaluation is largely ad hoc. Consequently, while XAI has the potential to be useful, its reliability remains constrained by the lack of ways to accumulate progress.

# 3. Fundamental Challenges for XAI

Based on the insights from our LLM-driven literature survey and survey study with researchers and practitioners, we now outline four fundamental challenges for explainable AI research. Although the problems we highlight in this section are fundamental, our aim is not merely to critique the field. By clarifying why these challenges matter, we hope to provide a clearer starting point for approaching them. We offer recommendations and illustrative examples that sketch out possible first steps. We begin with the main open challenge that XAI is facing today — the lack of definitions.

### 3.1. Challenge 1: Lack of Definitions

The absence of clear and formal definitions has been a significant issue for XAI. It can be viewed on three levels:

**Definition of explanation.** Despite the extensive body of methods work in XAI (Das & Rad, 2020), there remains no consensus even on the most fundamental question: what constitutes an explanation? Can any type of information qualify as an explanation? How does it depend on use case and stakeholder needs? Here, it may be fruitful to draw on the literature from philosophy, psychology, and cognitive science, where the question of what constitutes an explanation has long been explored (Pitt, 1988; Miller, 2019; Alvarez-Melis et al., 2021).

**Definition of a specific type of explanation.** Even for a specific type of explanation, precise agreed-upon definitions are often missing. A prominent example is feature-based explanations (Dwivedi et al., 2023), for which there is no formal gold-standard definition. Similarly for

concept-based methods, the definition of a "concept" is underspecified (Elhage et al., 2022).

**Free parameters.** Many explanation methods introduce free parameters without specifying their semantic meaning, leaving it unclear how these parameters should be understood. For example, LIME (Ribeiro et al., 2016) depends on choices such as the neighborhood definition, but it is not explicit what aspect of an explanation this parameter is intended to control. A similar ambiguity arises in sparse autoencoders (SAEs), which require selecting a sparsity ratio, non-linearity, and architecture, even though the connection between these choices and the explanatory content of the resulting representations is not formally articulated (Gao et al., 2024).

The lack of definitional clarity has resulted in a wide array of methods for tackling the same problems (Arya et al., 2019; Han et al., 2022), often without a unifying principle. At times, approaches even contradict one another, leaving researchers uncertain about which to adopt or how to compare their effectiveness (Krishna et al., 2022). Our survey study reflects this fragmentation: among 34 responses, researchers reported 6 explanation classes (e.g., feature attribution, data attribution, mechanistic interpretability) spanning 40 distinct methods. Even within feature attribution, 24 mentions were split across 16 different methods, underscoring the lack of consensus. This clutter impedes the field's progress. The lack of clarity not only discourages new researchers from entering and contributing to the field but also makes it difficult for those already in the field to navigate its disorganized landscape. This also emerged in our survey: when asked "What challenges did you encounter with the implementation of XAI methods?", 23.5% reported "I did not know which XAI method to choose" and 26.5% reported "It was unclear how to choose the XAI method's parameters." In addition, 44.1% highlighted the challenge of understanding or explaining the XAI results to end users, a crucial issue, since explanations lose their value if they cannot themselves be understood.

**Recommendations.** When proposing explainability methods, researchers should clearly define the concrete problem they aim to address. The problem statement does not need to be purely mathematical, especially when subjective human judgments are involved, but it must be defined well enough to specify *what* is being explained and to *whom*. Some approaches do offer well-specified objectives. For example, counterfactual explanations define a concrete objective, such as identifying minimal changes needed to alter a prediction, with a mathematical formulation that can be adapted to different tasks or user needs. This example illustrates the value of articulating the explainability goal explicitly, so that methods can be evaluated and critiqued against a clear and testable target.

## 3.2. Challenge 2: Unspecified Explanation Properties

To reason about explainability, we must specify the functional properties an explanation is expected to satisfy. While *faithfulness* is typically necessary[4] — an explanation claiming to reflect a model must be consistent with the underlying model behavior (Jacovi & Goldberg, 2020; Dasgupta et al., 2022; Lyu et al., 2024) — it is rarely sufficient on its own. In practice, explanations are expected to satisfy additional, goal-dependent properties such as completeness, robustness, fairness, or computational efficiency. Crucially, these properties complement other practical requirements, such as usefulness or actionability, whose definitions are inherently dependent upon specific stakeholders, use cases, and goals.

A core difficulty is that these properties are often discussed in isolation, without a clear statement of which behavioral aspect of the model they are intended to capture or why that property matters for a given task. Faithfulness illustrates this problem. For common explanation classes such as saliency maps, multiple faithfulness tests have been proposed, including model randomization (Adebayo et al., 2018), perturbation-based tests (Samek et al., 2016), and retraining-based evaluations (Hooker et al., 2019). These tests operationalize different desiderata, and no single test captures all aspects of faithfulness. As a result, a method may pass some tests while failing others, leading to conflicting conclusions about whether it is "faithful" at all (Krishna et al., 2022). This ambiguity has contributed to longstanding disputes, such as whether attention constitutes an explanation (Jain & Wallace, 2019; Wiegreffe & Pinter, 2019). Similar issues arise for chain-of-thought explanations, where multiple, incompatible notions of faithfulness have been proposed (Atanasova et al., 2023; Turpin et al., 2023; Lanham et al., 2023).

In general, faithfulness alone does not determine usefulness. A counterfactual explanation, for instance, may be perfectly faithful to a model's decision boundary yet practically useless if it fails to provide actionable suggestions, such as modifying immutable features like age or race (Ustun et al., 2019; Keane et al., 2021). As another example, explanations that summarize model behavior often trade strict fidelity for improved usability. Whether these trade-offs are acceptable depends entirely on the downstream task. Without explicitly defining the required properties and intended purpose, one cannot meaningfully assess an explanation's quality.

Completeness provides a concrete example of this problem. We use completeness to mean whether an explanation covers enough of the relevant model behavior, evidence, or decision logic for its intended use. For example, a saliency

---

[4] Faithfulness may not be required, for instance, when an end user's goal is only to understand whether to rely on a particular model output (Kim et al., 2025).

map may correctly highlight a breast lesion in a mammogram that a radiologist had previously identified, yet fail to identify any risk factors, anything related to a diagnosis, or help to determine what action should follow (Barnett et al., 2021). Existing work formalizes completeness through conservation-style attribution properties (Sundararajan et al., 2017; Lundberg & Lee, 2017) or concept sufficiency for recovering model predictions (Yeh et al., 2020). However, the required form of completeness depends on the downstream task. Thus, well-defined explanation properties should specify what aspects of model behavior an explanation covers and why that scope is sufficient.

This lack of clarity is reflected in practice. In our survey, 44.1% of practitioners reported not knowing how to assess explanation quality, and evaluation practices varied widely, often relying on ad hoc sanity checks or expectations rather than principled criteria. These findings suggest that the central challenge is the absence of clearly articulated properties that the explanations are intended to satisfy.

**Recommendations**. Explanations should be measured by explicit behavioral targets and downstream goals. Research must specify required properties, manage inherent trade-offs, and align methods with intended use cases, e.g., prioritizing robustness for regulatory auditing versus computational efficiency for interactive debugging. Formalizing these alignments, shifts the evaluation of explanations from standalone representations to functional workflow components.

### 3.3. Challenge 3: Lack of Coherent Evaluation

The evaluation of explanations is currently fragmented, lacking a cohesive framework that connects mathematical objectives to real-world utility. We argue that a rigorous evaluation pipeline must address three interconnected levels:

**Property guarantees**: Given a precise definition of a property (e.g., faithfulness, robustness), we can evaluate whether a method provably satisfies the property under stated assumptions. This gets at whether the property holds mathematically, not whether the explanation is useful in practice. Such guarantees do not necessarily need to cover the entire learning pipeline or require a fully formal theory of deep learning. For example, counterfactual explanations can satisfy well-defined feasibility or sparsity guarantees under explicit assumptions even when the underlying predictive model remains complex.

**Functionally-grounded metrics:** We can define automated proxies that empirically measure explanation quality (e.g., sparsity, faithfulness scores, completeness scores) on real models and datasets but without human intervention.

**Empirical evaluation with humans:** We can validate whether an explanation assists stakeholders in downstream tasks. This requires real models, datasets, and stakeholders with concrete tasks or goals. Because these details vary by domain, we do not advocate a single universal benchmark. Rather, evaluations should explicitly specify the task, stakeholders, success criteria, and relevant trade-offs. Human-in-the-loop evaluation should measure whether explanations improve performance on such tasks, rather than whether users find them plausible, satisfying, or faithful. This distinction is especially important for generative models, whose explanations may sound coherent while failing to reflect the model's actual behavior. Such task-grounded evaluation provides strong evidence for explanatory utility, but is resource-intensive.

Current literature treats these categories as distinct silos rather than a continuous pipeline. Vertically, Doshi-Velez & Kim (2017) organize evaluation by human involvement: application-grounded (experts performing real-world tasks), human-grounded (laypeople performing simplified proxy tasks), and functionally-grounded (no human involvement). Horizontally, Coroama & Groza (2022) distinguish between the nature of the signal: subjective measures versus objective measures.

By failing to connect these levels, researchers often optimize objective metrics like faithfulness in a vacuum — treating them as functionally-grounded proxies — without verifying if they correlate with any application-grounded utility. This disconnect has created a "phantom" category of evaluation: metrics that are mathematically precise but practically meaningless. Our practitioner survey confirms this failure: while academia produces ample metrics, only 35.3% of practitioners actually use quantitative measures. Instead, nearly half (44.1%) report "not knowing how to check explanation quality" as a primary bottleneck. Practitioners ignore existing objective metrics because they lack proven relevance to their real-world workflows.

Rather than treating human effort and automated metrics as a dichotomy, we propose **task-grounded objective evaluation**, in which automated explanation metrics are explicitly validated as proxies for performance on a specific downstream task involving human decision-makers. In contrast to traditional functionally-grounded metrics, which assume properties like "stability" or "sparsity" are universally desirable, task-grounded evaluation treats these properties as hypotheses whose relevance must be empirically established for a given task. Machine learning researchers could collaborate with researchers in HCI and other human-centered fields to test these hypotheses and ensure that the proxies are valid. Prior work on human-centered evaluation and debugging workflows provides useful examples of this direction, including HIVE (Kim et al., 2022), explanation-driven debugging (Kim et al., 2023), concept-level debugging of prototype-based networks (Bontempelli et al., 2023), and personalized alignment of prototypical

parts (Michalski et al., 2025). Ultimately, the field requires objective measures of pragmatic success rather than explainability for its own sake. This does not fully solve the problem of grounding explanations in model internals, but it makes evaluation more concrete and falsifiable by tying proxy metrics to measurable downstream outcomes.

**Recommendations.** We advocate for evaluation frameworks that are both task-specific and precisely defined. Metrics should be explicitly justified by the downstream interventions they support, moving beyond "general-purpose" heuristics. Any evaluation should be situated within a binding task context and measured by unambiguous, reproducible metrics. Of course, task-grounded objective evaluations cannot fully replace empirical evaluation with humans in the loop, but rather serve as a less resource-intensive complement to them.

### 3.4. Challenge 4: Lack of Actionability

In many settings, explanations should support concrete downstream goals, though understanding what a model is doing may itself be the goal. When explanations are intended to support decisions, debugging, oversight, or feedback, they should be connected to the actions or judgments they are meant to enable. While this goal-oriented framing is standard within the HCI literature on XAI (Hong et al., 2020; Suresh et al., 2021; Langer et al., 2021; Liao & Vaughan, 2024), it is frequently absent within the machine learning literature. This disconnect is evident in our practitioner survey: 38.2% of respondents could not integrate explanations into workflows, and 44.1% struggled to communicate outputs to decision-makers.

The problem is that many methods stop at showing information about the model, without specifying how that information should be used, by whom, and for what purpose. Within the HCI community, there is a long tradition of viewing transparency and control as interconnected design goals (Amershi et al., 2014; Liao & Vaughan, 2024). Providing explanations without control can frustrate users, while meaningful control may not be achievable without explanations (Smith-Renner et al., 2020; Storms et al., 2022). However, most XAI methods do not specify a well-defined object of control. They reveal information about model behavior without formalizing what elements of the system can be modified, by whom, and under what constraints. As a result, explanation-driven intervention is ill-posed even when explanations themselves are informative. For instance, while a model developer may want to use explainability to modify parameters or training data, a domain expert may instead seek to enforce logical constraints (e.g., safety) without the authority to retrain the model.

Such structural deficiencies create a *translation gap* between stakeholder requirements (e.g., "have your LLM not output unsafe text") and the mechanical updates required to fix the model. This has been observed in SAE research, where SAE-based explanations were found to be inadequate in steering model behaviour, and underperform simpler baselines (Wu et al., 2025; Bhalla et al., 2024). Without formally modeling which actions are feasible and authorized for a given explanation, explanation-driven interventions become ill-posed, breaking model behavior elsewhere. This is also observed in human–AI interaction surveys, where systems that provide visibility into behavior also deny meaningful agency over it (Raees et al., 2024).

Closing this gap requires translating the design principles established in HCI into formal computational mechanisms within XAI. While the former has successfully articulated the need for control, the latter has yet to develop the necessary pipelines that explicitly model *intervention logic* and *feedback propagation*. Future work must formalize how an explanation serves as an interface for updates: defining exactly what object is modified (e.g., parameters vs. constraints), who is authorized to modify it, and how that feedback propagates through the system. Unlike simpler interactive ML loops (Fails & Olsen Jr, 2003; Kulesza et al., 2015; Teso & Kersting, 2019), XAI requires richer system-level mechanisms to handle complex, high-dimensional feedback. Developing such mechanisms is a necessary next step to transform explanations from passive observations into active tools for actionability and alignment.

**Recommendations.** We advocate for explainability pipelines that are designed with actionability in mind. Such systems must be role-aware, i.e., know who their stakeholders are, and support iterative updates. Frameworks must establish role-aware interaction mechanisms that map stakeholder feedback to valid explanation or model updates. Rather than remaining static, explanations should be embedded within interfaces that allow users to specify desired reasoning, flag problematic behavior, or request modifications to a model's decision process. Even lightweight mechanisms can be effective. For example, collections of diverse counterfactuals (e.g., DiCE, Mothilal et al., 2020) enable stakeholders to select decision logic aligned with their goals. Such mechanisms demonstrate how explanations can move from observation to intervention and refinement.

## 4. A Call to Action: Foundations-driven XAI

### 4.1. What Becomes Possible with Foundational Work

Addressing the foundational challenges identified in Section 3 opens up new classes of research questions that are currently difficult to study in a principled way. These opportunities span how explanations are used by stakeholders, inform model evaluation, or integrate into data science workflows, and how progress in XAI can be measured cumu-

| 1. CLEAR DEFINITION | 2. FALSIFIABLE CLAIMS | 3. FAITHFULNESS | 4. APPLICATION | 5. EVALUATION IN CONTEXT |
|---|---|---|---|---|
| ✅ Precise objective ❌ Circular or undefined goals | ✅ Claims are testable ❌ Axiomatic claims with no evidence | ✅ Clear objective and test ❌ Visualization as validation | ✅ Clear real-world use case ❌ Vague claims like "helps trust" | ✅ Tested in intended real setting with users ❌ Only synthetic metrics |

*Figure 2.* A five-point checklist for evaluating explainability papers. For each criterion, green checkmarks indicate desirable properties, while red crosses illustrate common pitfalls that undermine rigor or clarity.

latively. With precise definitions of explanations and their desired properties, researchers can move beyond anecdotal case studies and begin to design task-specific benchmarks, controlled user studies, and formal evaluation protocols that directly test the utility of explanations for debugging, auditing, or decision support. For example, explanations could be treated as objects in model evaluation, enabling comparisons between models that exhibit similar predictive performance but differ in their reasoning patterns. Currently, such analyses are often designed for one failure type at a time, such as spurious correlations or rare subgroup errors. More systematic protocols would help compare models by whether they rely on appropriate evidence, not only by accuracy.

### 4.2. Explainability Paper Checklist

To help resolve the foundational challenges we discussed in this paper, we provide practical recommendations in terms of an explainability checklist (see Figure 2). Each criterion below is grounded in existing work: while no single paper may satisfy all criteria simultaneously, prior research demonstrates that each of them is achievable in isolation. Our intent is to make these practices explicit and encourage their systematic integration. Progress on these challenges also requires review processes to recognize problem formulation, evaluation design, and human-centered system integration as substantive contributions, not merely as supporting material for new methods.

**1. Definitions**: Does the paper provide a technically precise definition of its explainability sub-goal, grounded in the intended use and stakeholders? An **illustrative example** is counterfactual explanation research, which defines explanations through a precise objective of identifying minimal, feasible changes to the input required to alter a model's prediction (Wachter et al., 2018; Ustun et al., 2019; Mothilal et al., 2020). A common **negative example**, is extracting "human-understandable concepts" from models, without formalizing what constitutes a concept (Dwivedi et al., 2023; Elhage et al., 2022).

**2. Falsifiable Claims**: Are the paper's explainability claims falsifiable? That is, does there exist an experiment that could refute the main claim? Influential critiques in XAI serve as **useful examples** by designing experiments that could invalidate their hypotheses, such as sanity checks (Adebayo et al., 2018). Prior studies also discuss falsifiability, arguing

that evaluation is often conducted via anecdotes or proxy assumptions rather than falsifiable tests (Doshi-Velez & Kim, 2017; Nauta et al., 2023). In particular, Nauta et al. (2023) highlight that among the papers they analyzed, "1 in 3 papers evaluate exclusively with anecdotal evidence."

**3. Explanation Faithfulness**: Does the paper evaluate the faithfulness of explanations to actual model behavior, using tests aligned with the stated explanatory objective? **Illustrative examples** evaluate faithfulness using multiple, complementary tests rather than relying on a single visualization or proxy metric, such as benchmark-based evaluations for saliency methods (Hooker et al., 2019; Dasgupta et al., 2022) and recent work that formalizes and empirically tests faithfulness in natural language and chain-of-thought explanations (Atanasova et al., 2023; Lanham et al., 2023; Turpin et al., 2023). We cite these works as examples of making faithfulness claims testable, not as evidence that chain-of-thought explanations are generally faithful.

**4. Practical Applicability**: Does the paper describe a concrete real-world application of the explainability sub-goal? Research and applications of actionable recourse provides **useful examples** of how explanations can support decision-making by explicitly modeling feasible interventions and constraints (Ustun et al., 2019; Keane et al., 2021).

**5. Task-Specific Human-in-the-loop Evaluation**: Does the paper evaluate the explanation method in its intended real-world setting with humans in the loop? When this is not possible, validated task-grounded evaluations can be used. Some of the **positive examples** are related to model auditing (Gao et al., 2023; Metzen et al., 2023; Liang et al., 2023; Ribeiro & Lundberg, 2022; Yu et al., 2024). Demonstrating such applications provides perhaps the strongest "gold-standard" evidence for explainability.

### Acknowledgments

We thank Bingqing Chen for contributions and discussions toward an early version of the LLM-driven literature survey. We thank Sunnie Kim for helpful discussions and feedback on the manuscript, including pointers to relevant work on human-centered evaluation and debugging. We also thank Chhavi Yadav for feedback on the survey of researchers and practitioners and assistance with its outreach.

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

# A. LLM-driven Literature Survey

**Objective of LLM Survey.** We do not view LLM surveys as a replacement for human-driven literature surveys. While expert human researchers can understand the nuances of a field and make abstract connections between concepts, we do not believe that LLMs are, as of today, capable of this. For this reason, we view the role of the LLM as characterizing the claims made by papers, rather than the scientific merit of the claims themselves. For example, LLMs may be unable to ascertain whether a NeurIPS paper proposes a method that truly explains a black-box model, or whether the evaluations truly evaluate faithfulness in a meaningful manner. However, they can be used to determine whether a NeurIPS paper *claims* to propose an explainability method or *claims* to evaluate faithfulness, which is what this survey focuses on. Furthermore, given the lack of formal definitions for explainability, the only way to determine whether a paper studies explainability is whether the paper claims that it does.

*Table 1.* Summary of LLM-driven Literature Survey Results. Fractions in the table represent the fraction of papers evaluated as "True" for a given question and a given conference. We observe that the high-level trends are remarkably stable across conferences, indicated by the small standard deviation values in the last column.

| ID | Question | NeurIPS23 | NeurIPS24 | ICML23 | ICML24 | ICLR24 | ICLR25 | Avg | Std |
|---|---|---|---|---|---|---|---|---|---|
| 1 | Does this paper claim to primarily study post-hoc explanations of black-box machine learning models? Answer true if it claims to be primarily about post-hoc explanations, false otherwise. | 0.2517 | 0.1667 | 0.2692 | 0.2569 | 0.2927 | 0.2843 | 0.2536 | 0.0453 |
| 2 | Does this paper claim to primarily study machine learning models that are interpretable-by-construction? Answer true if it claims to be primarily about interpretable-by-construction models, false otherwise. | 0.4834 | 0.5000 | 0.5577 | 0.4679 | 0.5041 | 0.4902 | 0.5006 | 0.0308 |
| 3 | Does this paper claim to be primarily about mathematical theory for interpretability / explainability? Answer true if it claims to be primarily regarding theory for interpretability / explainability, and false otherwise. | 0.2368 | 0.1667 | 0.1579 | 0.3086 | 0.1413 | 0.2086 | 0.2033 | 0.0624 |
| 4 | Does this paper claim to propose a novel method for interpretability / explainability as its main contribution; or does it primarily perform analysis on existing methods? Answer true if it claims to propose a novel method as its primary contribution, false otherwise. | 0.7544 | 0.8016 | 0.7632 | 0.7531 | 0.7500 | 0.7975 | 0.7700 | 0.0234 |
| 5 | Does this paper claim to study interactive explanations, or static one-shot explanations? Answer true if it claims to study interactive explanations, false otherwise. | 0.1404 | 0.1587 | 0.2632 | 0.1358 | 0.2065 | 0.2025 | 0.1845 | 0.0490 |
| 6 | Does the paper claim to contain a formal mathematical definition for what either interpretability or explainability entails? Answer true if it claims to contain a mathematical definition, false otherwise. | 0.0702 | 0.0714 | 0.1579 | 0.0988 | 0.1413 | 0.1104 | 0.1083 | 0.0359 |
| 7 | Does the paper claim to demonstrate any concrete downstream impact of interpretability or explainability via experiments? Some examples of concrete downstream impacts include, debugging models, improving model performance or identifying spurious correlations. | 0.7544 | 0.7063 | 0.8421 | 0.7037 | 0.7935 | 0.7791 | 0.7632 | 0.0534 |
| 8 | Does this paper claim to demonstrate downstream applications in computer vision tasks? For our purpose, vision language models perform computer vision tasks. | 0.5789 | 0.4365 | 0.4474 | 0.4691 | 0.3804 | 0.4356 | 0.4580 | 0.0661 |
| 9 | Does this paper claim to demonstrate downstream applications in pure natural language processing tasks? For our purpose, vision language models do not perform NLP tasks. | 0.2281 | 0.2302 | 0.1053 | 0.3210 | 0.3152 | 0.3436 | 0.2572 | 0.0889 |
| 10 | Does this paper claim to demonstrate downstream applications in speech recognition? | 0.0088 | 0.0000 | 0.0263 | 0.0000 | 0.0000 | 0.0061 | 0.0069 | 0.0102 |
| 11 | Does the paper claim to propose a new evaluation metric for evaluating interpretability / explainability claims? Answer true if it claims to propose a new evaluation metric, false otherwise. | 0.1930 | 0.1587 | 0.1316 | 0.1481 | 0.1630 | 0.2393 | 0.1723 | 0.0386 |
| 12 | Does the paper claim to conduct a user study with real humans for evaluating interpretability / explainability benefits? Answer true if it claims to conduct a user study, false otherwise. | 0.1140 | 0.1111 | 0.1053 | 0.0864 | 0.1087 | 0.1104 | 0.1060 | 0.0100 |
| 13 | Does the paper claim to conduct a simulation-based evaluation for simulating a user study to evaluate interpretability benefits? Answer true if it claims to conduct a simulation-based evaluation, false otherwise. | 0.0439 | 0.0397 | 0.0263 | 0.0247 | 0.0435 | 0.0613 | 0.0399 | 0.0134 |

| ID | Question | NeurIPS23 | NeurIPS24 | ICML23 | ICML24 | ICLR24 | ICLR25 | Avg | Std |
|---|---|---|---|---|---|---|---|---|---|
| 14 | Does the paper claim to conduct experiments to validate the faithfulness of explanations? Answer true if it claims to conduct faithfulness experiments, false otherwise. | 0.4912 | 0.5159 | 0.5789 | 0.5309 | 0.6304 | 0.6074 | 0.5591 | 0.0549 |
| 15 | Does the paper claim to compare and benchmark against existing interpretability or explainability methods in the literature? Or does it claim to introduce a novel approach completely distinct from existing paradigms? Answer true if it compares and benchmarks, false otherwise. | 0.7368 | 0.6905 | 0.8158 | 0.7407 | 0.6957 | 0.7669 | 0.7411 | 0.0467 |
| 16 | Does the paper claim to be primarily about feature attribution or saliency maps? Answer true if it claims to be about feature attribution or saliency maps, false otherwise. | 0.1842 | 0.1111 | 0.2368 | 0.2469 | 0.1522 | 0.1534 | 0.1808 | 0.0528 |
| 17 | Does the paper claim to be primarily about concept-based explanations, including dictionary learning and Sparse Autoencoders? Answer true if it claims to be about concept-based explanations, false otherwise. | 0.3158 | 0.3254 | 0.3421 | 0.2593 | 0.2826 | 0.3374 | 0.3104 | 0.0328 |
| 18 | Does the paper claim to be primarily about data attribution? | 0.0175 | 0.0159 | 0.0526 | 0.0123 | 0.0435 | 0.0184 | 0.0267 | 0.0169 |
| 19 | Does the paper claim to be primarily about mechanistic interpretability? | 0.1770 | 0.2339 | 0.0789 | 0.1728 | 0.1957 | 0.3006 | 0.1932 | 0.0734 |
| 20 | Does the paper claim to primarily study interpretability or explainability of models that are not neural networks or transformers? Answer true if it studies models that are not neural networks or transformers, false otherwise. | 0.2018 | 0.2063 | 0.2895 | 0.1235 | 0.0761 | 0.0920 | 0.1649 | 0.0819 |

As a robustness check, we also applied the same LLM-based protocol to related papers from CVPR 2025 and ACL 2025. We analyzed 79 of 2871 accepted CVPR papers and 108 of 1978 accepted ACL papers that were identified as related to interpretability or explainability (see Table 2). This additional analysis is not intended as an exhaustive survey in all communities, but as a check on whether the high-level trends observed in NeurIPS, ICML, and ICLR are specific to those venues.

Overall, the main trends remain similar across venues: most papers propose new methods, while relatively few provide formal mathematical definitions or conduct human-centered evaluations. At the same time, the adjacent venues exhibit expected domain-specific differences. CVPR and ACL papers more often demonstrate concrete downstream applications, reflecting their applied focus, while formal theory and formal definitions are less common. ACL contains a larger fraction of mechanistic interpretability papers than CVPR, consistent with the stronger presence of mechanistic analyses of language models in NLP venues. These results suggest that the foundational issues identified in Section 2.1 are not unique to the ML venues in our main survey, although their manifestations vary by community and application domain.

## B. Survey Study with Researchers and Practitioners

In this section we provide additional information relating to our survey of researchers and practitioners. Table 3 describes the demographics of the survey respondents, and Table 4 reports the full survey questions and the responses we obtained.

Of the 43 respondents, 18 (41.9%) reported using inherently interpretable models. These divide into two groups. Nine respondents (20.9%) used *only* interpretable models; we refer to them as the interpretable ML subgroup. The remaining nine used interpretable models alongside post-hoc XAI techniques and therefore fall within the broader XAI population (N=34). In other words, the XAI respondents also drew on inherently interpretable models—26.5% of them did so—but the interpretable ML subgroup used no post-hoc explanation methods at all, reporting none of the post-hoc method classes in Table 4.

For this reason, throughout the main paper we report results over the XAI population (N=34), excluding the nine interpretable ML only respondents: because that subgroup used no post-hoc methods, their answers to questions about XAI method choice, evaluation, and challenges do not reflect experience with the techniques those questions concern. Tables 3 and 4 nonetheless report all three populations (the full sample, the XAI subgroup, and the interpretable ML subgroup) for completeness.

This difference in method use shows up in two questions in particular. For "How did you evaluate the correctness of the explanations?", the interpretable ML respondents either noted explicitly that the model was inherently interpretable, or

*Table 2.* Additional LLM-driven survey results for adjacent venues. The ML Conferences column reports the mean and standard deviation across the NeurIPS, ICML, and ICLR venues analyzed in the main survey. CVPR 2025 and ACL 2025 results are reported for related papers identified in those venues.

| Question | ML Conferences | CVPR 2025 | ACL 2025 |
|---|---|---|---|
| Q1: Post-hoc explanations | $0.25 \pm 0.05$ | 0.149 | 0.312 |
| Q2: Interpretable-by-construction | $0.50 \pm 0.03$ | 0.383 | 0.299 |
| Q3: Mathematical theory | $0.20 \pm 0.06$ | 0.038 | 0.018 |
| Q4: Novel method | $0.77 \pm 0.02$ | 0.769 | 0.661 |
| Q5: Interactive explanations | $0.18 \pm 0.05$ | 0.231 | 0.196 |
| Q6: Formal mathematical definition | $0.11 \pm 0.04$ | 0.000 | 0.036 |
| Q7: Concrete downstream impact | $0.76 \pm 0.05$ | 0.846 | 0.857 |
| Q8: Computer vision applications | $0.46 \pm 0.07$ | 1.000 | 0.125 |
| Q9: NLP applications | $0.26 \pm 0.09$ | 0.000 | 0.768 |
| Q10: Speech applications | $0.01 \pm 0.01$ | 0.000 | 0.000 |
| Q11: New evaluation metric | $0.17 \pm 0.04$ | 0.115 | 0.196 |
| Q12: User study | $0.11 \pm 0.01$ | 0.192 | 0.196 |
| Q13: Simulation-based evaluation | $0.04 \pm 0.01$ | 0.077 | 0.143 |
| Q14: Faithfulness validation | $0.56 \pm 0.05$ | 0.615 | 0.554 |
| Q15: Benchmarks existing methods | $0.74 \pm 0.05$ | 0.654 | 0.643 |
| Q16: Feature attribution / saliency | $0.18 \pm 0.05$ | 0.231 | 0.089 |
| Q17: Concept-based explanations | $0.31 \pm 0.03$ | 0.423 | 0.250 |
| Q18: Data attribution | $0.03 \pm 0.02$ | 0.000 | 0.036 |
| Q19: Mechanistic interpretability | $0.19 \pm 0.07$ | 0.077 | 0.268 |
| Q20: Non-neural-network models | $0.16 \pm 0.08$ | 0.000 | 0.000 |

appear to have interpreted the question as concerning usefulness for domain experts (citing user studies, expert-knowledge comparisons, or domain-specific sanity checks); notably, none of them reported using a quantitative metric. This pattern is consistent with the fact that faithfulness is given by construction for inherently interpretable models, so these practitioners did not face the post-hoc evaluation problem of verifying that an explanation reflects the model. For "To what extent did the XAI method help in achieving your goals?", 77.8% (7 of 9) rated "Helped significantly" and the remaining 22.2% (2 of 9) rated "Helped moderately," placing every respondent in the top two satisfaction categories.

*Table 3.* Information about our survey study participants.

| Question | Answer | Population (Rate ↓) | | |
|---|---|---|---|---|
| | | All (N=43) | XAI (N=34) | Interp. ML (N=9) |
| Affiliation | Academia | 67.4% | 70.6% | 55.6% |
| | Industry | 20.9% | 17.6% | 33.3% |
| | Research Institute | 4.6% | 5.9% | 0% |
| | Standards Organization | 2.3% | 2.9% | 0% |
| | Government Agency | 2.3% | 0% | 11.1% |
| | Unspecified | 2.3% | 2.9% | 0% |
| Years of experience | 9+ | 30.2% | 26.5% | 44.4% |
| | 3 − 6 | 27.9% | 26.5% | 33.3% |
| | 6 − 9 | 20.9% | 20.6% | 22.2% |
| | 1 − 3 | 20.9% | 26.5% | 0% |
| Research area | Computer Science and related fields | 72.5% | 67.7% | 88.9% |
| | Other fields | 20% | 22.6% | 11.1% |
| | Healthcare and Medicine | 7.5% | 9.7% | 0% |

# C. Open questions and problems in the field

## C.1. Selected Open Problems for Future Research

Rather than attempting an exhaustive enumeration of all gaps in the field, we highlight a few specific open problems where foundational progress would be particularly helpful for research. Drawing from our analysis of the challenges in Section 3, we identify the following questions as a good starting point for advancing the science of explainability.

*Table 4.* Our survey study results. Rate refers to the fraction of respondents in each population who chose the respective answer. All questions permitted multiple responses.

| Question | Answer | Population (Rate ↓) | | |
|---|---|---|---|---|
| | | **All** (N=43) | **XAI** (N=34) | **Interp. ML** (N=9) |
| Which class of XAI method(s) did you use / study in that project? | Feature attribution methods | 55.8% | 70.6% | 0% |
| | Inherently interpretable model | 41.9% | 26.5% | 100% |
| | Concept-Based Explanations | 23.3% | 29.4% | 0% |
| | Counterfactual Explanations | 20.9% | 26.5% | 0% |
| | LLM-based Explanations | 18.6% | 23.5% | 0% |
| | Mechanistic Interpretability | 9.3% | 11.8% | 0% |
| | Data attribution | 7% | 8.8% | 0% |
| Within the chosen class of XAI methods, why did you choose the specific method(s) you chose? | Easy to implement / had good document | 53.5% | 52.9% | 55.6% |
| | It is a popular method | 46.5% | 47.1% | 44.4% |
| | Backed by rigorous research and testing | 37.2% | 29.4% | 66.7% |
| | User-friendly interface for the end-user | 25.6% | 23.5% | 33.3% |
| Who was the intended audience for the explanations? | ML Researchers and Practitioners | 60.5% | 61.8% | 55.6% |
| | Non-ML Domain Experts | 55.8% | 50.0% | 77.8% |
| | Non-experts | 25.6% | 29.4% | 11.1% |
| What was the intended goal of the explanations? | Understanding patterns in datasets | 65.1% | 58.8% | 88.9% |
| | Enhancing trust | 62.8% | 58.8% | 77.8% |
| | Debugging and Model improvement | 53.5% | 52.9% | 55.6% |
| | Compliance with regulations | 9.3% | 8.8% | 11.1% |
| To what extent did the XAI method help in achieving your goals? | Helped significantly | 51.2% | 44.1% | 77.8% |
| | Helped moderately | 30.2% | 32.4% | 22.2% |
| | Helped slightly | 11.6% | 14.7% | 0% |
| | Not applicable | 4.7% | 5.9% | 0% |
| | Didn't help at all | 2.3% | 2.9% | 0% |
| How did you evaluate the correctness of the explanations? | Compared the explanations with expert knowledge | 46.5% | 44.1% | 55.6% |
| | Conducted sanity checks relevant to the problem domain | 39.5% | 38.2% | 44.4% |
| | Used a clear quantitative metric | 27.9% | 35.3% | 0% |
| | User study | 16.3% | 11.8% | 33.3% |
| | Did not evaluate explanation correctness, but I expect them to be correct because they were consistent with my expectations | 11.6% | 14.7% | 0% |
| | Did not evaluate explanation correctness, and not sure if they were correct | 7% | 8.8% | 0% |
| What challenges did you encounter with the implementation of XAI methods? | I did not know how to check the explanation quality | 41.9% | 44.1% | 33.3% |
| | It was challenging to understand or explain the XAI results to end users | 41.9% | 44.1% | 33.3% |
| | I encountered challenges integrating the XAI methods into my existing workflow or system | 37.2% | 38.2% | 33.3% |
| | It was unclear how to choose the XAI method's parameters | 27.9% | 26.5% | 33.3% |
| | I faced challenges with the computational resources or data needed to implement the XAI methods | 23.3% | 29.4% | 0% |
| | I did not know which XAI method to choose | 20.9% | 23.5% | 11.1% |
| | I did not find an easy off-the-shelf implementation | 11.6% | 11.8% | 0% |

1. The "holy grail": How should we formally define explanations?
   And a step down in difficulty: How can we mathematically define a specific class of explanations (e.g., concept explanations or feature attributions)?

2. Perhaps the simplest (or most practical): How can we systematically select the free parameters in XAI algorithms (e.g., perturbation type in LIME, sparsity ratio in SAEs, . . . )? How do we interpret their meaning?

3. How can we design "gold standard" datasets and benchmarks that accurately measure progress in XAI research? In other words, what is the "Imagenet" for XAI research?

4. Can explainability methods help with model validation for real-world deployment, or characterization of model behavior? Is explainability necessary for validation?

5. How can we design end-to-end pipelines that connect explanation, human feedback, and model/explanation edits to enable efficient interaction with stakeholders?

