# OpenReview forum: "Position: Explainability Research Must Prioritize Foundations over Ad-hoc Methods"
_ICML.cc/2026/Position_Paper_Track — ICML 2026 Position Paper Track regular_

### Official Review · Reviewer_KHmw · 2026-03-11

**Significance:** 4
**Argument Clarity:** 4
**Rating:** 5
**Confidence:** 5

**Questions:**

I do not have question in relation to the position itself. Minor suggestion to broader the spectrum of XAI related work outside three conference and include works listed before or venues for human-computer interactions, e.g. "Help me Help AI" work

**Alternative Views Section:**

Yes

**Compliance With Llm Reviewing Policy A Conservative:**

Affirmed.

**Discussion Potential:**

3

**Final Justification:**

I've taken into consideration the rebuttal. I will increase my score to acknowledge the chances, however I am still not fully convinced about the exclusion of other communities.

**Paper Summary:**

The paper put a thesis that XAI research stuck. And is lacking fundamental definitions, benchamarking and purpose. There is no agreement on ontology within XAI, no agreement on properties that XAI method should have, that users are often neglected in evaluation of XAI. Most of them also lack application definition and evaluation in the context. Those points are discussed further within the work to highlight its importance, and specific call to actions are made to improve over those dimensions. E.g. discussion on faithfullness and completeness. It proposes the use of XAI work checklist and human in the loop evaluation.

**Position:**

Yes

**Position In Title:**

Yes

**Related Work:**

2

**Strengths And Weaknesses:**

Position is well stated, it is clearly presented and explained. However it is limited to works from NeurIPS, ICML and ICLR, which may miss work from other domains, e.g. computer vision where some of the arguments are being addresses, but maybe not sufficiently.

The position is well supported, and the topic is of the most importance for ICML community, I would even say for broader computer science community and in interdisciplinary research where XAI holds a promise of identification of novel knowledge, e.g. in drug discovery and biology.

Definitely, it will inspire discussion.

Most of the citations is relevant and they are quite extensive.

There is no discussion about XAI evaluation by humans done already in the community [1] or in debugging sessions [2, 3]

[1] Kim, Sunnie SY, et al. "HIVE: Evaluating the human interpretability of visual explanations." European Conference on Computer Vision. Cham: Springer Nature Switzerland, 2022.

[2] Bontempelli, Andrea, et al. "Concept-level debugging of part-prototype networks." ICLR 2023.

[3] Michalski, Tomasz, et al. "Personalized Interpretability--Interactive Alignment of Prototypical Parts Networks." arXiv preprint arXiv:2506.05533 (2025).

**Support:**

4

---

> ### Author Rebuttal · Authors · 2026-03-31
>
> Thank you very much for your review and feedback on our paper!
>
> We agree that interpretability and explainability work in adjacent communities, including computer vision, is highly relevant here. Our empirical discussion focused on ML venues such as NeurIPS, ICML, and ICLR because these are the conferences where much of the general-purpose XAI methodology discussed in the paper is published. Our goal was therefore not to suggest that the issues we identify are unique to these conferences, but to keep the scope bounded and aligned with the ICML audience. We also note that our practitioner survey included researchers from diverse backgrounds and application areas, and their reported challenges align closely with the issues identified in our literature survey, suggesting these problems are not limited to any single community.
>
> We thank the reviewer for pointing out HIVE (Kim et al., 2022) “Help Me Help the AI” (Kim et al., CHI 2023), concept-level debugging (Bontempelli et al., 2023), and personalized interpretability (Michalski et al., 2025). We will add citations to all of these works in the revision to strengthen the paper's discussion of human-centered and task-grounded evaluation.
>
> Thank you again for helping us improve the clarity of our paper! We are happy to discuss any points during the discussion period.

---

> > ### Author Rebuttal · Reviewer_KHmw · 2026-04-02
> >
> > I understand the point of focusing only on 3 conferences, but I still believe this work would benefit from broader space, including CV and language conferences as right now all of those areas are highly intersecting. That is why I am not raising my score.

---

### Official Review · Reviewer_rjLW · 2026-03-16

**Significance:** 4
**Argument Clarity:** 3
**Rating:** 6
**Confidence:** 4

**Questions:**

1. How many papers were surveyed in section 2.1?
2. What does “precisely defined” mean in the context of explanation evaluation? It would be a good idea to present an example.

**Alternative Views Section:**

Yes

**Compliance With Llm Reviewing Policy A Conservative:**

Affirmed.

**Discussion Potential:**

4

**Paper Summary:**

This paper argues that the current research does not address the issue of integrating explanations into end-to-end, human-in-the-loop systems. It further argues that the research on XAI should shift from developing ad-hoc methods to addressing the foundational and structural challenges. Based on this argument, this paper presents some findings about current XAI research and a survey from XAI researchers. This paper further includes a checklist for XAI research practice to help shift XAI toward a more human-centered, action-oriented paradigm.

**Position:**

Yes

**Position In Title:**

Yes

**Related Work:**

4

**Strengths And Weaknesses:**

**Strengths**

This paper did a good job on several aspects.

- First of all, the topic of this position is certainly timely and important. XAI research is now in the moment that reflections should be done and also do some forward-looking based on the what have happened. The issue pointed out by this paper is certainly one of the major issues in XAI research: how can we make XAI more useful.
- The work conducted in this research is solid and informative. For example, a review about recent XAI publications echoed the issue pointed out by this paper. The survey from XAI researchers and practitioners provided further evidence for the argument.
- As a position paper, this paper also provided a practical guideline for future work on XAI. I enjoyed reading section 4.2, even though I may not fully agree with some of the points.

**Weaknesses**

- Certain recommendations can be further clarified, as the one listed in the question part.

**Support:**

4

---

> ### Author Rebuttal · Authors · 2026-03-31
>
> Thank you for your thoughtful and encouraging review! We’re glad that you liked our position, and we hope the discussion can help us improve our arguments.
>
> ---
>
> > How many papers were surveyed in section 2.1?
>
> We surveyed 617 papers that were determined as papers relating to interpretability or explainability according to Gemini 2.5. This was across 6 major conferences - ICML 23/24, NeurIPS 23/24, ICLR 24/25.
>
> ---
>
> > What does “precisely defined” mean in the context of explanation evaluation? It would be a good idea to present an example.
>
> Great question! To present a negative example of an evaluation that is not precisely defined, saliency research often uses pixel perturbation as a faithfulness evaluation, where the idea is to remove unimportant pixels and measure change in model outputs. However the manner of pixel “removal”, replacing with zeros or data mean, is often considered a free hyper-parameter, and its choice can in practice lead to significant divergence in performance [1]. We can add this example in the main paper. To present a recent (prospective) positive example, [2] present a mechanistic interpretability evaluation based upon secret elicitation from language models, which is well-defined as it does not appear to contain such critical free hyper-parameters.
>
> [1] Tomsett et al., “Sanity Checks for Saliency Metrics”, AAAI 2020
>
> [2] Cywinski et al, “Eliciting Secret Knowledge from Language Models”, arxiv 2025

---

### Official Review · Reviewer_LeX3 · 2026-03-22

**Significance:** 2
**Argument Clarity:** 3
**Rating:** 3
**Confidence:** 3

**Questions:**

see Weaknesses.

**Alternative Views Section:**

Yes

**Compliance With Llm Reviewing Policy A Conservative:**

Affirmed.

**Discussion Potential:**

3

**Final Justification:**

Some of my points were addressed, but my concerns around artificially narrowing the scope of explainability while claiming a broad position, by excluding certain subfields of explainable AI research such as Mechanistic Interpretability, have not been addressed. Specifically, the authors did not respond to my latest response and recommendation to narrow the scope, as not all of explainability research is about stakeholder utility, while the authors claim it is based on self-identification of an LLM-survey.

**Paper Summary:**

The paper advocates for introducing rigor and structure into the research field on Explainable AI (XAI).
The problem is that current research on XAI is widely scattered and unstructured, oftentimes based on anecdotal evidence and not utility oriented.
To support this view, they review recent surveys on XAI and provide and complement their assessment with a user study that aligns well with the made points that the field is primarily focused on methods, essential task-grounded evaluations are lacking, and majority of study participants is unclear on how to check explanation quality.
Based on their assessment they outline four challenges, namely establishing definitions, unspecified explanation properties, lack of coherent evaluations, and lack of actionability along with recommendations how to tackle each of them.
Finally, they provide a checklist for future work on XAI that comprises five key points: definitions, falsifiable claims, explanation faithfulness, practical applicability, and human-in-the-loop evaluation.

**Position:**

Yes

**Position In Title:**

No

**Related Work:**

3

**Strengths And Weaknesses:**

**Strengths:**

- I like the call for interdisciplinary collaborations, I am convinced it is needed for AI researcher to be able to tackle the really hard problems of our time.
- The authors provide a comprehensive review of the state of XAI research including a user survey and LLM-assisted filtering (validated for human aggreement)
- The challenges of research on XAI are identified make sense
- The checklist for future works on XAI is actionable and useful
- The paper is well written and easy to follow

**Weaknesses:**

- My main concern is that the position of the authors clearly values engineering utility over scientific discovery, which effectively disregards XAI research that uncovers the inner functions of deep learning models as non-valuable unless it serves real-world use cases. However, discovery of inner workings of current models may contribute to a better understanding and potentially theoretical insights and derivations.
- The human-in-the-loop gold standard: I question whether this is the correct approach, given that humans nowadays frequently anthropomorphize reasoning traces [1] and can easily be fooled by capable LLMs [2]. This makes evaluation based on humans-in-the-loop brittle and purely objective task-agnostic evaluation metrics may still capture such effects, rendering them useful afterall.
- The title is a bit misleading, specifically the term "foundations" can be interpreted as mathematical first-principles instead of the HCI-centric foundations the authors refer to. Effectively this shifts the definition of "foundations" to "functional utility", a title along the lines of "Explainability Research Must Prioritize Human Utility over Exploratory Interpretability." would improve clarity.
- The call for "provable property guarantees" (line 303, right) is a goal that, in the absence of a unifying theory of deep learning, forces a trade-off where rigor can only be achieved by using low-complexity, non-competitive models.
-The shift toward "task-grounded objective evaluation" (line 103, right) does not actually move beyond heuristics, it simply replaces general-purpose proxy metrics with narrower, task-specific ones that still lack a ground-truth link to model internals.
- To me it does not make sense to use CoT as an example for "Explanation Faithfulness", the space of language is combinatorially infinitely large, hence there will always be an example that will falsify the faithfulness of a CoT.

**References:**

[1] Stop anthropomorphizing intermediate tokens as reasoning/thinking traces!, Kambhampati et al., 2024

[2] Language Models Don't Always Say What They Think: Unfaithful Explanations in Chain-of-Thought Prompting, Turpin et al., NeurIPS 2023

**Support:**

3

---

> ### Author Rebuttal · Authors · 2026-03-31
>
> We thank the reviewer for their careful reading and constructive feedback!
> - **W1: The position of the authors clearly values engineering utility over scientific discovery.**
> Our position is not that scientific discovery is less valuable than engineering utility. Our paper targets a specific category of work: methods that explicitly claim to produce explanations useful for stakeholders in real-world high-stakes decision support. Research that investigates the inner workings of deep learning models as a scientific pursuit, without claiming stakeholder utility, is a different category and is not the target of our critique. We will add the following clarifying sentence in the introduction: “Our argument primarily targets methods that claim explanatory utility for stakeholders, and is not intended to discourage research that investigates model internals to advance deep learning theory and understanding.”
> - **W2: Is human-in-the-loop gold standard the correct approach, given that humans can be fooled by capable LLMs?**
> We agree that humans can be fooled by plausible-sounding explanations and can anthropomorphize reasoning traces. This is precisely why we do not advocate subjective human judgment of explanation quality as the gold standard. Instead, our proposal is task-grounded evaluation: measuring whether an explanation helps a human perform better on a concrete, measurable task, such as identifying model failures, deciding when to override a prediction, debugging or correcting the system, or making a more accurate downstream decision. This differs from asking whether an explanation seems faithful or plausible, which is indeed brittle.
> For the same reason, purely objective task-agnostic metrics are not sufficient on their own. Such metrics are only meaningful to the extent that they are validated against outcomes that matter in the intended use case. We therefore view objective metrics as proxies whose value depends on whether they predict performance on the relevant human task, rather than as substitutes for task-grounded evaluation. We will clarify in the paper that our human-in-the-loop standard refers to human task performance, not subjective judgments of explanation quality.
> - **W3: The term foundations can be interpreted as mathematical first-principles instead of HCI-centric foundations.**
> The term "foundations" is intentionally chosen by analogy to mathematical first principles: just as mathematical foundations specify the basic assumptions and building blocks on which later results depend, we use the term to refer to the underlying principles that should guide XAI methods intended to support human decision-making. In our view, human utility is part of these foundations and should be treated as a design requirement throughout the development process, rather than as an afterthought introduced only at evaluation or application time. We will revise the abstract to make the intended meaning of foundations and their human-centric requirements more explicit.
> - **W4: Provable property guarantees force a trade-off where rigor requires non-competitive models.**
> We are not asking for end-to-end formal guarantees over full deep learning pipelines. We are asking for a more modest goal: when a paper claims an explanation satisfies a property, that claim should be testable and falsifiable under explicitly stated assumptions. Counterfactual explanations are a concrete example, where one can prove sparsity or feasibility properties without needing a general theory of deep learning. We will add a clarifying sentence at line 303.
> - **W5: Task-grounded objective evaluation does not move beyond heuristics.**
> We agree that task-grounded evaluation does not resolve ground-truth links to model internals, which remains an open problem. However, existing proxy metrics evaluate properties such as stability in isolation, without validating whether they matter for any externally measurable outcome. Task-grounded evaluation anchors evaluation to concrete task performance instead. We will clarify in the revision that task-grounded evaluation is a step toward making this problem more concrete and testable, not a complete solution.
> - **W6: CoT should not be used as an example for Explanation Faithfulness.**
> We do not claim CoT is a faithful explanation. The cited work, including Turpin et al. 2023 and Lanham et al. 2023, is used as an example of how to test and formalize faithfulness. In fact, Turpin et al. 2023 explicitly demonstrates that CoT can be unfaithful. Our point is that these works make faithfulness claims falsifiable and testable. We will add a clarifying sentence in Section 4.2.
>
>
> Thank you again for helping us improve the clarity of our paper! We are happy to discuss any points during the discussion period.

---

> > ### Author Rebuttal · Reviewer_LeX3 · 2026-04-02
> >
> > Thank you for the elaborate responses, I agree with most statements made in the rebuttal.
> > I have one more remaining point, quoting the authors: "Our paper targets a specific category of work: methods that explicitly claim to produce explanations useful for stakeholders in real-world high-stakes decision support.", makes clear to me that they are targeting a sub-area of XAI, but the field of XAI also comprises research on scientific discovery and theoretical explanations that do not necessarily enhance utility for humans performing in high-stake domains. Therefore I recommend to clarify this and make the position not about the field of XAI in general, but about the specified sub-domain.

---

### Official Review · Reviewer_Wtkx · 2026-03-22

**Significance:** 3
**Argument Clarity:** 2
**Rating:** 4
**Confidence:** 3

**Questions:**

1. Section 2.2 finding 6 shows 81.4% of practitioners found XAI methods beneficial. How do you reconcile this with the claim that foundational shortcomings are the primary bottleneck? Can you quantify the gap between the current ad-hoc utility and the utility you project with stronger foundations?
2. Your LLM survey validated on 25 (paper, question) pairs at 88% agreement. What is the base rate that a greedy "True" vs "False" classifier achieves? If 90% are "False," a trivial classifier achieves 90%. Can you provide per-question-type accuracy and confidence intervals for the headline statistics?
3. Can you name a concrete, measurable criterion for "sufficient foundational progress", i.e., a condition under which method development should be re-prioritized?

**Alternative Views Section:**

Yes

**Compliance With Llm Reviewing Policy A Conservative:**

Affirmed.

**Discussion Potential:**

3

**Paper Summary:**

This paper argues that Explainable AI (XAI) research should be more grounded in a user-centric, actionable paradigm and implies that such a paradigm might increase its impact on real-world applications. They analyze recent XAI papers at ICML/NeurIPS/ICLR and conduct a practitioner survey. The paper identifies four interdependent challenges (lack of definitions, underspecified explanation properties, bad evaluation, and lack of actionability) and proposes a five-point checklist to reorient the field.

**Position:**

Yes

**Position In Title:**

Yes

**Related Work:**

3

**Strengths And Weaknesses:**

### Strengths

- Section 1.1 acknowledges strong counterarguments to their own proposal, and makes a clear response to them.
- The four identified challenges (definitions, properties, evaluation, and actionability) have interesting logical dependencies, which are clearly stated.
  - The point that faithfulness is necessary but insufficient (faithful explanations can reference immutable features) is non-obvious and well-made.
  - The observation that only 27.9% of practitioners use quantitative metrics for evaluation, and that they ignore existing objective metrics because they lack proven relevance to real-world workflows, is also sharp.
- Section 4.2's five-point checklist is well-structured and actionable, especially because it includes positive and negative examples grounded in specific prior work.
- In addition to argumentation, the paper provides "empirical evidence" in the form of a literature survey and a practitioner survey.

### Weaknesses

- I'm not sure what to make of the fact that (line 223 left) 81.4% of surveyed practitioners found XAI methods beneficial. The paper treats this as evidence of "unrealized potential" but never quantifies the gap between current and potential utility.
- The paper's thesis, "Foundations should be prioritized over methods," specifies no measurable criterion for when foundations are sufficient. The checklist items are high-level enough that reasonable people will disagree on whether they are satisfied.
- The paper does not engage with the possibility that foundational consensus emerges *from* "ad-hoc" methods development (e.g., as happened in statistical learning theory after SVMs, or much of modern ML after deep learning). I think a clear counterargument against this stance in section 1.1 would be helpful for the paper.

**Support:**

2

---

> ### Author Rebuttal · Authors · 2026-03-31
>
> Thank you for your thoughtful and encouraging review! We’re glad that you liked our position, and we hope the discussion can help us improve our arguments.
>
> ---
>
> > Section 2.2 finding 6 shows 81.4% of practitioners found XAI methods beneficial. How do you reconcile this with the claim that foundational shortcomings are the primary bottleneck?
>
> To dispel a possible confusion, the unrealized potential claim is not based on the 81.4% figure alone. The stronger evidence comes from the pain points reported in the same survey: 41.9% of practitioners do not know how to check explanation quality, and 37.2% cannot integrate explanations into their workflows. Practitioners are extracting value under conditions that are, by their own account, poorly supported. However, the reviewer is right that we do not make this reasoning explicit, and we will revise the paper to clarify this point.
>
> ---
> > Your LLM survey validated on 25 (paper, question) pairs at 88% agreement. What is the base rate that a greedy "True" vs "False" classifier achieves? If 90% are "False," a trivial classifier achieves 90%. Can you provide per-question-type accuracy and confidence intervals for the headline statistics?
>
> Among the 25 QA pairs, 66% were False, while the remaining were True. Thus the LLM achieves a classification rate far above the greedy classifier. While we agree that per-question-type accuracy / confidence intervals would strengthen the results, obtaining these requires extensive human annotations which are costly to obtain. Nonetheless, we will revisit adding these to the final draft of the paper.
>
> ---
> > Can you name a concrete, measurable criterion for "sufficient foundational progress", i.e., a condition under which method development should be re-prioritized?
>
> Great question! We would like to reference Thomas Kuhn’s “The structure of scientific revolutions” here, which describes science as alternating between two phases: Normal Science (stable puzzle-solving within a shared paradigm) and Revolutionary Science (a crisis-driven paradigm shift). Methods development is well suited to Normal Science, when foundations are well-established. We argue in this paper for rethinking foundations, i.e., a paradigm shift which emphasizes downstream utility. A concrete criterion is thus to establish good frameworks and benchmarks to hill-climb on, such that progress on these benchmarks meaningfully push the frontier of explainability. However, ultimately, we do not think there can be a “measurable criterion” here. In particular, Kuhn’s work argues that the establishment of a paradigm is often only known post-hoc after community agreement.
>
> ---
> > The paper does not engage with the possibility that foundational consensus emerges from "ad-hoc" methods development (e.g., as happened in statistical learning theory after SVMs, or much of modern ML after deep learning). I think a clear counterargument against this stance in section 1.1 would be helpful for the paper.
>
> Thank you for this comment! While we do discuss ad hoc methods development in section 1.1, we are happy to expand this discussion and address the argument explicitly. That said, we believe the historical analogy may not hold for XAI. In the SVM and deep learning cases, foundational consensus emerged after methods because they had real utility: predictive performance improved, benchmarks converged, and the empirical signal was clear. XAI is fundamentally different, the methods are not just theoretically ungrounded, but we lack a clear empirical signal of progress. We therefore believe that continuing to develop ad-hoc methods in XAI is more likely to compound the existing confusion than to resolve it.
>
> Thank you again for helping us improve the clarity of our paper! We are happy to discuss any points during the discussion period.

---

> > ### Author Rebuttal · Reviewer_Wtkx · 2026-04-05
> >
> > Thank you for the rebuttal. Most of my initial concerns and questions are addressed.
> >
> > I appreciate the clarification on my point vs deep learning and agree; the way I'm reading this is that "foundational progress" in the way the paper describes could enable rigorous evalution which in turn could enable similar progress. I'd like to lightly push back on the claim that “XAI is fundamentally different,” since the distinction may be less one of kind than of current evaluability. But aside from that wording, the rebuttal resolves my concern.
> >
> > The reference to Kuhn is useful context, but I still find the answer a bit non-operational for a paper making a field-level prioritization claim. While the response is intellectually/philosophically reasonable, the way I read the paper's central claim is as a practical and normative one about how the field should allocate effort; in my view, the argument remains difficult to translate into practical guidance for the subfield.

---

### Decision · Program_Chairs · 2026-04-30

**Decision:**

Accept (regular)

**Comment:**

This position paper raises major issues with the XAI field that is relevant and timely, especially at this point in time when a wide range of XAI techniques have been proposed in the literature. The arguments presented by the authors is very compelling and supported by extensive and informative research. The authors also include an additional analysis in their rebuttal that covers CVPR 2025 and ACL 2025. Reviewers also appreciated the concrete checklist in Section 4.2.

In terms of weaknesses, some reviewers found it difficult to draw practical, actionable guidance from the position argued. A reviewer felt the paper was too broad in its position scope and title as they felt the paper excluded certain subareas of XAI. Finally, the paper could also include the possibility that ad hoc methods could inform development of foundations as in other subareas of ML; the authors' response to this point was informative and should be incorporated into the paper.

Overall, the major strengths far outweigh the weaknesses.